# Association of Nursery School-Level Promotion of Vegetable Eating with Caregiver-Reported Vegetable Consumption Behaviours among Preschool Children: A Multilevel Analysis of Japanese Children

**DOI:** 10.3390/nu13072236

**Published:** 2021-06-29

**Authors:** Yukako Tani, Manami Ochi, Takeo Fujiwara

**Affiliations:** 1Department of Global Health Promotion, Tokyo Medical and Dental University, Tokyo 113-8519, Japan; ochi.m.aa@niph.go.jp (M.O.); fujiwara.hlth@tmd.ac.jp (T.F.); 2Department of Health and Welfare Services, National Institute of Public Health, Saitama 351-0104, Japan

**Keywords:** eating pattern, eating vegetables first, eating behaviour, preschool children, school level, Japan

## Abstract

Nursery schools can play an important role in children developing healthy eating behaviours, including vegetable consumption. However, the effect of school-level vegetable promotion on vegetable consumption and body mass index (BMI) remains unclear. This study examined the associations of nursery school-level promotion of eating vegetables first at meals with Japanese children’s vegetable consumption behaviours and BMI. We used cross-sectional data collected in 2015, 2016, and 2017 on 7402 children in classes of 3–5-year-olds in all 133 licensed nursery schools in Adachi, Tokyo, Japan. Caregivers were surveyed on their children’s eating behaviours (frequency of eating vegetables, willingness to eat vegetables and number of kinds of vegetables eaten), height and weight. Nursery school-level promotion of eating vegetables first at meals was assessed using individual responses, with the percentage of caregivers reporting that their children ate vegetables first at meals as a proxy for the school-level penetration of the promotion of vegetable eating. Multilevel analyses were conducted to investigate the associations of school-level vegetable-eating promotion with vegetable consumption behaviours and BMI. Children in schools that were 1 interquartile range higher on vegetable promotion ate vegetable dishes more often (β = 0.04; 95% CI: 0.004–0.07), and were more often willing to eat vegetables (adjusted odds ratio = 1.17; 95% CI: 1.07–1.28), as well as to eat more kinds of vegetables (adjusted odds ratio = 1.19 times; 95% CI: 1.06–1.34). School-level vegetable-eating promotion was not associated with BMI. The school-level health strategy of eating vegetables first may be effective in increasing children’s vegetable intake but not in preventing being overweight.

## 1. Introduction

The prevalence of children being overweight or obese has increased worldwide in recent decades [1], now being 25% in Organization for Economic Cooperation and Development countries [2]. Vegetables are an important component of a healthy diet, and low vegetable intake has been linked to increased risk of diet-related diseases, including obesity [3]. In 2019, however, the average daily vegetable intake for Japanese adults was 281 g, and only about 30% consumed the recommended 350 g per day [4]. There is no internationally consistent recommended vegetable intake; however, health authorities such as the World Health Organization/Food and Agriculture Organization recommend a daily intake of ≥400 g/day of fruit and vegetables, which corresponds to five or more servings of 80 g/day [5]. Many countries add to this recommendation that three of the five servings should comprise vegetables (≥240 g/day) and two fruit (≥160 g/day) [6]. As it has been reported that the frequency of fruit intake in Japan is less than half of that in European countries [7], the recommended intake of vegetables for Japanese should likely be set higher than specified by the World Health Organization. A recent systematic review estimated that increasing vegetable intake to 300 g per day could potentially decrease the risk of most associated diseases by 10% to 30% [8]. Therefore, increasing vegetable intake to the recommended level has the potential to reduce the burden of some diet-related diseases [8]. As eating patterns formed early in life track into adolescence and adulthood [9,10,11], childhood is a critical period for establishing the habit of eating vegetables. Additionally, interventions may be more successful for younger children than for older children [12].

It is not easy to make eating vegetables habitual for children, as they innately prefer sweet and salty foods and reject sour and bitter foods [13]. Particularly for young children, the bitter taste of vegetables is a major barrier to vegetable consumption [14]. Internationally, a high proportion of children have an inadequate fruit and vegetable intake [15]. A study among pre-school children showed that only 17% of them consumed the recommended five or more servings of fruit and vegetables [16]. In Japan, the average daily vegetable intake of children aged 1–6 years was 129 g in 2019, which is well below the recommended levels [4,17]. Various strategies have been implemented to increase children’s vegetable intake, including nutrition education programs, parental involvement and experiential learning (i.e., school gardens, in-school cooking and food preparation activities) [18,19]. A review of the research on children’s eating behaviour found that the experiential learning approach may be more effective compared with nutrition education programs for increasing children’s vegetable intake [19]. However, the experiential learning approach requires time, as well as financial and human resources. In a survey of nursery school teachers in Japan, “events” were particularly burdensome for nursery school teachers, and 90% of these teachers felt that their jobs included a lot of work other than childcare [20]. Given the finite resources, there is a need to identify a time- and effort-saving strategy for increasing children’s vegetable intake.

A health promotion strategy known as “eating vegetables first at meals” was launched in 2013 in Adachi, Tokyo, with the aim of developing healthy eating habits [21]. The municipality mainly focused on children aged 4 and 5 years attending nursery schools, and nursery school teachers promoted eating vegetables as the first bite at meals. The purpose of the present study was to examine the associations of nursery school-level vegetable-eating promotion with vegetable consumption behaviours and BMI among Japanese children.

## 2. Materials and Methods

### 2.1. Study Population

We used cross-sectional survey data on eating behaviours and BMI among preschool children collected in Adachi, Tokyo, Japan. The survey, which aimed to investigate children’s eating behaviours as part of a diabetes prevention program implemented in Adachi, was conducted in 39 nursery schools in November 2015, 51 nursery schools in December 2016 and 51 nursery schools in November 2017 (Appendix A). In 2015, we invited all public-build and operate nursery schools among the licensed nursery schools. In 2016 and 2017, we recruited all public-build and private operate nursery schools, in addition to public-build and operate nursery schools. All recruited nursery schools participated in the survey. The nursery school principals explained the aim and details of the survey to the caregivers of 3–5-year-old children. A questionnaire on eating behaviour, body height and body weight was distributed to 9231 caregivers of 3–5-year-old children (2651 in 2015, 3362 in 2016, and 3218 in 2017). A total of 7970 (2307 in 2015, 2898 in 2016 and 2765 in 2017) returned the questionnaire (response rate: 86%) (Appendix A and Appendix A.) Questionnaires with missing information on a child’s sex (*n* = 121), failure to complete the questions related to a child’s eating behaviours (*n* = 194) or failure to provide a child’s body height, body weight or month of birth (*n* = 253) were excluded from the analysis. After these exclusions, 7402 participants (2304 in 2015, 2897 in 2016 and 2765 in 2017) were included in the analysis. The survey was conducted anonymously, and no personally identifiable identification variable was included, so the same children may be included in multiple survey years (Appendix A). The mean number of children per school was 56.9 (standard deviation [SD] = 15.6). The survey was conducted by Adachi City, and the use of the secondary data was approved by the Institutional Review Board of Tokyo Medical and Dental University (approval number: M2016-284-02). Participants were informed that participation in the study was voluntary, and that completing and returning the questionnaire indicated their consent to participate.

### 2.2. Children’s Eating Behaviours and Body Weight Status

Children’s eating behaviours, body height and body weight were assessed by a caregiver using a questionnaire. To assess children’s eating behaviours related to vegetable intake, we evaluated the frequency of eating vegetable dishes and two behaviours related to neophobia that interfere with children’s vegetable consumption [14]. Children’s frequency of eating vegetable dishes was assessed using the question “How often does your child eat vegetable dishes? Vegetable dishes include, for example, salads, side dishes of boiled seasoned vegetables, simmered vegetables, miso soup or soups with vegetables. If your child eats vegetable dishes in their school lunch, count these in the number of dishes.” The response options were almost every meal, two meals a day and one meal a day or less. Two behaviours related to the lessening of neophobia were assessed: willingness to eat vegetables; and whether the child was eating more types of vegetables than they had eaten in the past. Children’s willingness to eat vegetables was assessed using the question “Does your child try vegetables on his/her own?” with the response options of yes and no. Whether the child ate an increasing variety of vegetables was assessed using the question “Is the variety of vegetables that your child can eat increasing?” with the response options of yes and no. Experts on Japanese preschool children’s diets selected two behaviours related to neophobia (content validity) on the basis of their feasibility in many nursery schools. Children’s height and weight were reported by caregivers via a questionnaire. Each child’s BMI was calculated by dividing the child’s weight (in kilograms) by the square of their height (in meters). BMI was expressed as a z-score representing the deviation in SD units from the mean of an age- and sex-specific standard normal distribution of BMI, according to the World Health Organization’s Child Growth Standards. Children’s BMIs were categorised as underweight/normal weight (<+1 SD) or overweight (≥+1 SD) using SD cut-offs.

### 2.3. School-Level Promotion of Eating Vegetables First

To assess school-level vegetable-eating promotion, in each year we calculated the proportion of children eating vegetables first at meals by nursery school (Appendix A). Our study sample (*n* = 7402) was nested within 133 nursery schools (31 nursery schools in 2015, 51 nursery schools in 2016 and 51 nursery schools in 2017). The average number of children per nursery school included in the sample was 59 in 2015, 57 in 2016 and 55 in 2017 (Appendix A). Eating vegetables first was assessed at the individual level using caregivers’ responses to the question “Does your child eat vegetables as the first bite at meals?” with the response options of yes and no. The proportion of caregivers responding *yes* for each nursery school was calculated and used as a proxy for the penetration of the vegetable-eating promotion. The interquartile range was calculated for analysis.

### 2.4. Covariates

Child’s sex (boy or girl), class (for 3-, 4- or 5-year-olds), family economic status (high, middle, low or unknown), caregiver’s age (<30, 30–39, 40–49 or ≥50 years) and caregiver’s nutritional knowledge (knew or did not know the recommended vegetable intake) were assessed using the caregivers’ responses to the questionnaire. More specifically, family economic status was classified as good, normal, poor (difficulty meeting living expenses) and unknown. Caregiver’s nutritional knowledge was assessed using the question “Do you know that the recommended vegetable intake is 350 g or more per day?” with the response options of yes and no. The recommended vegetable intake, 350 g, is based on the estimated vegetable consumption required to achieve the proper intake of potassium, dietary fiber and antioxidant vitamins [22].

### 2.5. Statistical Analysis

A multilevel model approach was used in this analysis. First, we estimated three-level hierarchical linear/logistic regression models. Level 1 indicators included all outcomes and characteristics of participants at the individual level. Level 2 indicators were the variables for survey year, and the school-level variable was the Level 3 indicator. The overall fit of the models was evaluated using the likelihood-ratio test. The models with individual- and survey year-level variables showed a non-significant difference in the likelihood-ratio test, compared with the models with only individual-level covariates. In contrast, the models with individual- and school-level variables showed a significant difference in the likelihood-ratio test, compared with the models with only individual-level covariates (*p* < 0.01). Therefore, we ultimately decided to estimate two-level hierarchical linear/logistic regression models: Level 1 included individual-level variables; and Level 2 included the school-level variable. Random intercepts and fixed-slopes models were used to calculate coefficients or multi-level odds ratios (ORs) and 95% confidence intervals (CIs). The models were adjusted for potential confounders, including the child’s sex, age and family economic status, as well as the caregiver’s age and nutritional knowledge as individual-level covariates. All analyses were conducted using Stata, Version 15.

## 3. Results

Table 1 shows the characteristics of the study sample. Around half of the children were girls, 34% were in the class for 4-year-olds and 35% were in the class for 5-year-olds. Nearly 60% of the households had high to middle overall economic status. The caregivers were mostly aged in their 30s and 40s (55.6% and 27.3%, respectively). A total of 62% of the caregivers reported knowing the child’s recommended daily vegetable consumption. The children’s mean frequency of eating vegetable dishes was 2.3 times per day, while 69% of the children tried vegetables on their own and 86% had increased the variety of vegetables they ate. Overweight children accounted for 13% of the sample.

The mean school-level percentage of children eating vegetables first was 35.5% (SD = 10.5). The minimum percentage of children eating vegetables first was 0%, and the maximum percentage of children eating vegetables first was 59.3%. The interquartile range was 15.3.

Table 2 shows the results of the multilevel models. We found that school-level promotion of eating vegetables first was significantly associated with better eating behaviours related to vegetable intake among children. Children enrolled in nursery schools where a larger proportion of children ate vegetables first at meals tended to eat more vegetable dishes (β = 0.04, 95% CI: 0.004–0.07), to be more willing to eat vegetables on their own (OR = 1.17, 95% CI: 1.07–1.28) and to have increased the variety of vegetables they ate (OR = 1.19, 95% CI: 1.06–1.34) compared with those enrolled in nursery schools, where a smaller proportion of children ate vegetables first at meals. School-level vegetable-eating promotion was not associated with children’s BMI. In terms of individual-level factors, being a girl, being older, having a higher household economic status, having a caregiver aged <40 years and having a caregiver who knew the recommended daily vegetable consumption were associated with better eating behaviours related to children’s vegetable intake. The odds of trying vegetables on one’s own were 1.5 times (95% CI: 1.35–1.66) higher for girls than for boys. The odds of the variety of vegetables eaten having increased were 1.3 times (95% CI: 1.11–1.54) higher for children in the class for 5-year-olds than for children in the class for 3-year-olds. The frequency of eating vegetable dishes was 0.18 (95% CI: 0.10–0.25) higher in households with a high economic status than in households with a middle economic status.

## 4. Discussion

To our knowledge, the present study is the first to examine the association between school-level vegetable-eating promotion and children’s eating behaviours. We found that school-level promotion of eating vegetables first at meals was associated with more frequent vegetable consumption, greater willingness to eat vegetables and more variety in vegetable intake for children, but no association was found between this school-level variable and BMI. The present study suggests that a simple intervention (i.e., encouraging children to eat vegetables as the first bite at meals as a default option) may lead to increased vegetable intake among preschool children in Japan.

School-level promotion of eating vegetables first at meals was associated with better eating behaviours related to vegetable intake among children. This finding is consistent with the result of a previous study examining the effect of the individual-level variable of eating vegetables first at meals on diet [23]. A study among Japanese preschool children showed that children who more often ate vegetables first at meals consumed 148 g of vegetables per 1000 kcal, which was much higher than the 68.0 g of vegetables per 1000 kcal consumed by children who rarely or never ate vegetables first at meals [23]. In our study, we found that school-level promotion of eating vegetables first at meals was associated with higher willingness to eat vegetables and an increase in the kinds of vegetables children would eat. Creating an environment where everyone eats vegetables first at meals may help children to overcome their neophobia, and promote their acceptance of vegetables.

The presence of peers may play an important role in promoting children’s acceptance of vegetables [19]. When children observe peers choosing vegetables, they shift their behaviour to also select their previously nonpreferred vegetables [24]. Although a detailed dietary survey was not conducted in our study, there is reason to suspect that promoting eating vegetables first in schools may lead to an increase in children’s vegetable intake. Because increasing numbers of women work outside the home in Japan [25], many children spend most working days in nursery schools from an early age. Thus, nursery schools can play an important role in establishing healthy eating habits among children [26].

A potential explanation for the association between school-level promotion of eating vegetables first at meals and children’s eating behaviours is “nudging”. Nudging has been defined as “any aspect of the choice architecture that alters people’s behaviour in a predictable way, without forbidding any options or significantly changing their economic incentives” [27]. One way to nudge vegetable intake involves the order in which food is eaten (i.e., the default option). A crossover study conducted among children in the United States found that serving vegetables or vegetable-based soup as a first course was associated with increased total vegetable consumption over the meal [28,29]. Our results are consistent with those of these previous studies. Therefore, encouraging children to eat vegetables first at meals may be effective in establishing the habit of eating vegetables, eating more vegetables or eating more types of vegetables.

School-level promotion of eating vegetables first at meals was not associated with children’s BMI. One potential reason for this finding is that it takes more time to change weight status than to change eating behaviours. Because our study had a cross-sectional design, we could not follow up individuals. Future study is necessary to examine the long-term outcomes and the continuity of the behaviour of eating vegetables first at meals.

A recent Cochrane review reported that it is uncertain whether parental nutritional interventions are effective in increasing the vegetable consumption of children under the age of 5 years [30]. Parents influence their children’s eating behaviours in various ways; to encourage desirable eating behaviours in their children, it is important that parents do not exert excessive control, but rather serve as good role models for eating and adopt an authoritative style of feeding [26]. In Japan, a typical meal consists of a staple food, a soup and three dishes (one main side dish and two side dishes) served all at once [31]. Therefore, the order of food consumption at a meal can vary significantly among individuals. Parents eating vegetables first at meals may be effective for encouraging children to eat vegetables in Japanese settings.

Among other individual-level factors, the acceptance of vegetables (i.e., increased willingness to eat vegetables by themselves and increased number of types of vegetables they would eat) was relatively high for girls, older children and children with a high economic status; these results are consistent with previous studies [32,33,34]. When we examined the interaction effects between school-level promotion of eating vegetables first and these individual factors, we found a significant interaction with child’s age on willingness to eat vegetables: the positive association between school-level promotion of eating vegetables first on willingness to eat vegetables was particularly prominent among older children (*p* < 0.05 for interaction). This finding suggests that older children benefit more from school-level promotion of eating vegetables first, which is intuitive as older children have been exposed to encouragement to eat vegetables first for a longer period of time, compared with younger children.

There are several limitations to our study. First, because caregivers reported their children’s eating behaviours, there may be misclassification, especially for school meals. However, caregivers could know how often their children ate vegetable dishes at school based on reports of the nursery school’s daily menu. Second, children’s eating behaviours were assessed using a very simple questionnaire that has not been validated. Considering the impracticability of large-scale research, this is a common limitation in this field. A caregiver’s social desirability may bias their child’s eating behaviours, possibly leading to underestimation of the association between promotion of school-level vegetable-eating and children’s eating behaviours. Third, children’s heights and weights were reported by caregivers via a questionnaire. A systematic review has shown that caregivers tend to underestimate the weight of overweight children [35]. Therefore, we may have underestimated the association of school-level promotion of eating vegetables first at meals with being overweight. However, nursery schools perform regular health checkups, including height and weight measurements, and report the results to the children’s caregivers. Fourth, the school-level promotion of eating vegetables first at meals may have been underestimated in this study, as the penetration of the promotion was calculated based on caregiver perceptions rather than the promotion activities implemented in nursery schools. However, the school-level contextual effect could be assessed with the measurement used in this study, and this is sufficient for multilevel analysis. Fifth, because the participants were selected from one city in Tokyo, generalizability is not high. Sixth, we could not eliminate the effects of potential confounding factors such as the caregiver’s education and presence of the child’s siblings. Although we did account for the caregiver’s nutritional knowledge, operationalised as their knowledge of the recommended daily vegetable consumption, the question related to caregiver’s nutritional knowledge was “suggestive”, and the response could depend on access to current policy information, which could have biased our findings. Future studies should use a questionnaire that has been established as valid and reliable. Additionally, some potentially confounding school-level factors, such as area deprivation, should be considered. Seventh, the recruitment of some children into a study multiple times could introduce substantial bias. However, we considered that the percentage of such cases would be equally distributed across the nurseries, as the participation grades and years were the same. Finally, because our study was cross-sectional in nature, reverse causation is likely, as our assessment of school-level vegetable-eating promotion was calculated using caregivers’ perceptions of whether their children ate vegetables first at meals.

## 5. Conclusions

This large-scale study has provided novel findings regarding the associations of school-level promotion of eating vegetables first with eating behaviours related to vegetable intake among preschool children in Japan. The current study may indicate the possibility of intervening in eating behaviours by encouraging children in school settings to eat vegetables first at meals. Further longitudinal studies are needed to clarify the causal relationships and long-term effects of eating vegetables first.

## Figures and Tables

**Table 1 nutrients-13-02236-t001:** Characteristics of enrolled preschool children and caregivers (n = 7402).

		*n*	%
Children’s characteristics			
Sex			
	Boy	3798	51.3
	Girl	3604	48.7
Age (class)			
	3-year-olds class	2251	30.4
	4-year-olds class	2539	34.3
	5-year-olds class	2612	35.3
Household status			
Economic status			
	High	369	5.0
	Middle	3846	52.0
	Low	894	12.1
	Unknown/missing	2293	31.0
Caregiver’s age			
	<30 years	1061	14.3
	30–39 years	4113	55.6
	40–49 years	2018	27.3
	≥50 years	78	1.1
	Missing	132	1.8
Caregiver’s knowledge of recommended daily vegetable consumption			
	Did not know	2787	37.7
	Knew	4577	61.8
	Missing	38	0.5
Survey year			
	2015	2083	28.1
	2016	2717	36.7
	2017	2602	35.2
Children’s weight status			
	Normal weight (BMI z-score < 1 SD)	6423	86.8
	Overweight (BMI z-score ≥ 1 SD)	979	13.2
Children’s eating behaviours			
Willingness to eat vegetables			
	Yes	5096	68.8
	No	2306	31.2
Whether the child ate an increasing variety of vegetables			
	Yes	6368	86.0
	No	1034	14.0
		Mean	SD
Frequency of eating vegetable dishes (number of times per day)		2.26	0.71

BMI: body mass index; SD: standard deviation.

**Table 2 nutrients-13-02236-t002:** Association of school-level percentage of children eating vegetables first at meals with children’s eating behaviours and BMI (n = 7402).

	Frequency of Eating Vegetable Dishes	Willingness to Eat Vegetables	Whether the Child Ate an Increasing Variety of Vegetables	Overweight (BMI Z-Score ≥ 1 SD)	BMI Z-Score
	β (95% CI)	OR (95% CI)	OR (95% CI)	OR (95% CI)	β (95% CI)
School-level factor					
Ate vegetables first at meals (IQR)	**0.04 (0.004–0.07)**	**1.17 (1.07–1.28)**	**1.19 (1.06–1.34)**	1.08 (0.96–1.23)	0.04 (−0.002–0.09)
Individual-level factors					
Child’s sex					
Boy	reference	reference	reference	reference	reference
Girl	**0.07 (0.03–0.10)**	**1.50 (1.35–1.66)**	**1.15 (1.01–1.31)**	**0.74 (0.64–0.85)**	−0.03 (−0.07–0.02)
Child’s age (class)					
3-year-olds class	reference	reference	reference	reference	reference
4-year-olds class	0.02 (−0.02–0.06)	**1.16 (1.02–1.31)**	**1.24 (1.06–1.46)**	0.98 (0.83–1.16)	−0.02 (−0.08–0.04)
5-year-olds class	**0.04 (0.004–0.08)**	**1.36 (1.20–1.54)**	**1.31 (1.11–1.54)**	0.88 (0.74–1.04)	**−0.08 (−0.14–−0.03)**
Economic status					
High	**0.18 (0.10–0.25)**	**1.33 (1.04–1.70)**	1.25 (0.89–1.75)	1.19 (0.87–1.62)	−0.02 (−0.12–0.09)
Middle	reference	reference	reference	reference	reference
Low	**−0.14 (−0.19–−0.09)**	**0.80 (0.68–0.93)**	**0.69 (0.56–0.83)**	1.22 (0.99–1.51)	−0.0002 (−0.07–0.07)
Unknown/missing	**−0.10 (−0.14–−0.07**)	1.12 (0.99–1.25)	1.04 (0.89–1.22)	1.17 (1.00–1.37)	0.05 (−0.002–0.10)
Caregiver’s age					
<30 years	0.01 (−0.04–0.06)	1.12 (0.96–1.31)	**1.59 (1.27–2.00)**	**0.79 (0.63–0.98)**	**−0.09 (−0.16–−0.02)**
30–39 years	ref	ref	ref	ref	ref
40–49 years	−0.04 (−0.07–0.00)	**0.81 (0.72–0.91)**	**0.86 (0.74–1.00)**	1.05 (0.90–1.23)	0.00 (−0.05–0.06)
≥50 years	**−0.40 (−0.55–−0.24)**	**0.53 (0.33–0.83)**	**0.41 (0.24–0.68)**	**2.05 (1.20–3.49)**	**0.27 (0.04–0.49)**
Missing	**−0.16 (−0.28–−0.04)**	0.86 (0.59–1.25)	0.75 (0.48–1.18)	1.13 (0.69–1.84)	0.02 (−0.15–0.19)
Caregiver’s knowledge of recommended daily vegetable consumption					
Did not know	ref	ref	ref	ref	ref
Knew	**0.08 (0.05–0.11)**	1.06 (0.95–1.17)	**1.31 (1.14–1.50)**	**1.19 (1.03–1.37)**	**0.09 (0.04–0.14)**
Missing	**0.27 (0.05–0.50)**	1.72 (0.78–3.79)	1.71 (0.60–4.88)	1.12 (0.43–2.90)	−0.09 (−0.41–0.23)
Random parameters					
School-level					
Variance (standard error)	0.005 (0.002)	0.023 (0.012)	0.030 (0.018)	0.051 (0.022)	0.012 (0.004)
Intra-class correlation (%)	1.0	0.7	0.9	1.5	1.2

BMI: body mass index; OR: adjusted odds ratio; CI: confidence interval. Boldface indicates statistical significance (*p* < 0.05).

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
