# Peer review of "Association of Nursery School-Level Promotion of Vegetable Eating with Caregiver-Reported Vegetable Consumption Behaviours among Preschool Children: A Multilevel Analysis of Japanese Children"

_nutrients, 2021, doi:10.3390/nu13072236_

Round 1

Reviewer 1 Report

The study aims to evaluate the association of nursery school-level promotion of vegetable eating with vegetable consumption behaviors and BMI. Although the methodology has some important limitations, the findings are relevant to assist in the understanding of behaviors that leverage the consumption of vegetables among the target population of the research. The sample is expressive and apparently representative of the target audience of the study. The authors clearly point out all the limitations of the study and indicated the necessary referrals to complement the results presented. However, some aspects can be considered to improve the presentation of the findings.

  1. As one of the main limitations indicated by the authors was the data collection based on the information provided by the caregivers, the title and method could indicate that the data is based on their perception. Thus, part of the limitations would be justified because the interest would be divided between the effect of Nursery school-level promotion of eating vegetables first at meals and the participation of caregivers.
  2. Regarding the sample, lines 78 - 82 show that “The survey,…, was conducted in 39 nursery schools in November 2015, 51 nursery schools in December 2016, and 51 nursery schools in November 2017…” Soon after, the authors reported that “ALL caregivers of 3–5-year-old children enrolled at ALL licensed nursery schools in Adachi were invited to participate in the survey.” Does the statement mean that all “nursey schools” in the city participated in the sample? Are there 51 nursey school in Adachi? Why the variation in the total number of schools per year? How was caregiver related to nursey school?
  3. Considering the cultural differences between countries, it is important to clarify the role of the caregiver in the education of Japanese children and the relationship among them.
  4. Another aspect is the time taken by each caregiver to return the questionnaire and whether each caregiver is responsible for only one child or more than one child.

In relation to the other questions, all were clarified on lines 246-274.

Author Response

We thank this reviewer for the helpful comments. We have carefully revised the manuscript accordingly.

  1. As one of the main limitations indicated by the authors was the data collection based on the information provided by the caregivers, the title and method could indicate that the data is based on their perception. Thus, part of the limitations would be justified because the interest would be divided between the effect of Nursery school-level promotion of eating vegetables first at meals and the participation of caregivers.

Reply:

Thank you for this important suggestion. Accordingly, we have added the following information to the title.

Title

“Association of nursery school-level promotion of vegetable eating with caregiver-reported vegetable consumption behaviors among preschool children: A multilevel analysis of Japanese children”

  1. Regarding the sample, lines 78 - 82 show that “The survey,…, was conducted in 39 nursery schools in November 2015, 51 nursery schools in December 2016, and 51 nursery schools in November 2017…” Soon after, the authors reported that “ALL caregivers of 3–5-year-old children enrolled at ALL licensed nursery schools in Adachi were invited to participate in the survey.” Does the statement mean that all “nursey schools” in the city participated in the sample? Are there 51 nursey school in Adachi? Why the variation in the total number of schools per year? How was caregiver related to nursey school?

Reply:

We apologize for our insufficient explanations. In 2015, we invited all public-build and operate nurseries among the licensed nurseries. In 2016 and 2017, we recruited all public-build and private operate nurseries in addition to public-build and operate nurseries among the licensed nurseries. Every year, all recruited nurseries participated in the survey. The total number of nursery school children changes from year to year because the number of enrolled children changes every year and the response rate also changes. Therefore, we have modified the method section as follows:

The survey, which aimed to investigate children’s eating behaviors as part of a diabetes prevention program implemented in Adachi, was conducted in 39 nursery schools in November 2015, 51 nursery schools in December 2016, and 51 nursery schools in November 2017 (Supplementary Figure 1). In 2015, we invited all public-build and operate nursery schools among the licensed nursery schools. In 2016 and 2017, we recruited all public-build and private operate nursery schools in addition to public-build and operate nursery schools. All recruited nursery schools participated in the survey. The nursery school principals explained the aim and details of the survey to the caregivers of 3–5-year-old children. A questionnaire on eating behavior, body height, and body weight was distributed to 9,231 caregivers of 3–5-year-old children (2,651 in 2015, 3,362 in 2016, and 3,218 in 2017).

  1. Considering the cultural differences between countries, it is important to clarify the role of the caregiver in the education of Japanese children and the relationship among them.

Reply:

Thank you for this important comment. Accordingly, we have added the following sentences to the Discussion section.

“A recent Cochrane review reported that it is uncertain whether parental nutritional interventions are effective in increasing the vegetable consumption of children under the age of 5 years [30]. Parents influence their children’s eating behaviors in various ways; to encourage desirable eating behaviors in their children, it is important that parents do not exert excessive control, but rather serve as good role models for eating and adopt an authoritative style of feeding [26]. In Japan, a typical meal consists of a staple food, a soup, and three dishes (one main side dish and two side dishes) served all at once.[31] Therefore, the order of food consumption at a meal can vary significantly among individuals. Parents eating vegetables first at meals may be effective for encouraging children to eat vegetables in Japanese setting.”

  1. Another aspect is the time taken by each caregiver to return the questionnaire and whether each caregiver is responsible for only one child or more than one child.

Reply:

Thank you for this comment. We have added the presence of siblings as potential confounding factors to the limitation as follows:

“Sixth, we could not eliminate the effects of potential confounding factors such as caregiver’s education and presence of child’s siblings.”

Reviewer 2 Report

The manuscript entitled “Association of nursery school-level promotion of vegetable eating with vegetable consumption behaviors and BMI: A multilevel analysis of Japanese children” presents interesting and important issue, however some minor corrections are needed.

  • The article is well written
  • Authors should think if the word "BMI" in tittle is necessary (this is only a suggestion).
  • Line 253 – “ Third, children’s heights and weights were reported by caregivers via questionnaire.” – please specify what type of bias it could be and how it could influence the results.  
  • All my previous comments were incorporated. Thank you. 

Author Response

We thank this reviewer for the helpful comments. We have carefully revised the manuscript accordingly.

  1. Authors should think if the word "BMI" in tittle is necessary (this is only a suggestion).

Reply:

Thank you for this important suggestion. Accordingly, we have deleted the “BMI” from the title.

  1. Line 253 – “ Third, children’s heights and weights were reported by caregivers via questionnaire.” – please specify what type of bias it could be and how it could influence the results.  

Reply:

Thank you for this very important comment. We have added the following sentence to the Discussion section.

“Third, children’s heights and weights were reported by caregivers via questionnaire. A systematic review has shown that caregivers tend to underestimate the weight of overweight children[35]. Therefore, we may have underestimated the association of school-level promotion of eating vegetables first at meals with overweight.”

This manuscript is a resubmission of an earlier submission. The following is a list of the peer review reports and author responses from that submission.

Round 1

Reviewer 1 Report

Major issues: 

  • The abstract section is much too long and its hard to follow. Many references and aspects described in there should be rather placed in the discussion rather than in the introduction. Moreover, some of those are irrelevant having in mind the aim of the study (Line #34-35 – impact of food environment on mortality and dementia in older people). The introduction section should provide clear rationale and background for what has been done in the study. In this case the introduction should answer/reflect following issues (max. 400-500 words):
    • Prevalence of obesity (including Japan); health burden (briefly!)
    • Role of vegetables intake in obesity prevention – current evidence
    • Gap in research indicating many possible strategies to promote vegetable intake – here drafting potential benefits of nursery schools
    • Aim of the study
  • The fact that children could be recruited into the study multiple times might introduce substantial bias especially when the percentage of such cases is not known.
  • Measurement of both vegetable intake as well as food neophobia using a unvalidated tool (question/screener in this case) causes serious risk of bias.
  • Likewise simplification of nutritional knowledge of caregiver to question about standard portion (which can even depend on access to the current policy information rather.
  • Different both dietary and non-dietary confounders could introduce substantial residual confounding (e.g. home food environment)
  • It is not mentioned whether child’s weight and height was measured by schools or it was self-reported by parents? This is also key information considering other potential biases.
  • What is the goal of presenting raw data in Supplementary Table 1?
  • Considering methodological weaknesses, I am sorry to say that I cannot refer to the results and discussion.

Minor issues:

  • The article does not follow the updated journal template.
  • References are not formatted according to the journal’s author guide
  • When interpreting the analyses results please consistently decide on interpretation of p-values or confidence intervals – e.g. (β = 0.04; 95% CI: 0.00–0.07) – report a proper number of decimal places or P-value – in its current presentation the CI contains null effect thus should not be regarded as present assocation.

Reviewer 2 Report

The manuscript entitled “Association of nursery school-level promotion of vegetable eating with vegetable consumption behaviors and BMI: A multilevel analysis of Japanese children” presents an important and interesting issues.

Introduction:

  • Line 34-39 – this paragraph is not associated with the main aim of the study. It should be deleted.
  • In this section Authors should present the information associated with the promotion of vegetable eating in school children. This section should be briefly presented – what do we know and what is the background for this study. Some detailed information about other studies are necessary (associated with the efficacy) . The good background should present the history of problem, the current knowledge and scientific "gap", and then authors should present how their study could fill this gap to justify the study.
  • Line 42 – some international contexts should be presented. Recommended by Ministry of Health Labour and Welfare (Japan) 350 g of vegetables per day is higher than recommended by WHO 400g of fruit and vegetables per day (excluding potatoes and other starchy tubers) - is about 240 g of vegetables per day and 160 g of fruits per day. Various countries have different recommendations – this aspect should be discussed.    

Material and methods section:

  • The applied questioners are not validated. This is some bias but it is indicated in the limitations section. However, it need to be addressed in the discussion (what can be the result of using not validated questioners)
  • Line 117 - Why only two behaviors related to neophobia were analyzed? Why authors did not use a Food Neophobia Scale? Please specify it.
  • Line 146 – “family economic status”. How the economic status was defined? Please specify it.
  • Line 150 “Do you know that the recommended vegetable intake is 350 g or more 150 per day?” -such type of question is “suggestive question” - how this may affect the obtained results

Results:

  • Table 2 is quite small – I think, that this data could be presented as a text. But this is just a suggestion.

Discussion:

  • This section should be boarded. Authors should compare gathered data with the results by other authors.

Round 2

Reviewer 2 Report

Authors made great effort to improve the manuscript, however some additional comments:

Line 25 – it should be “CI: 0.004–0.07” instead of “CI: 0.00–0.07”

Lines 80-86 – there is some typos

Lines 96-100 – the aim of the study was presented twice.

The manuscript should be carefully polish taking into account the typos.   
